# Controversies and Open Questions in Management of Cancer-Free Carriers of Germline Pathogenic Variants in *BRCA1/BRCA2*

**DOI:** 10.3390/cancers14194592

**Published:** 2022-09-22

**Authors:** Rinat Bernstein-Molho, Eitan Friedman, Ella Evron

**Affiliations:** 1The Oncogenetics Unit, Chaim Sheba Medical Center, Tel-Hashomer, The Sackler School of Medicine, Tel-Aviv University, Tel-Aviv 5265601, Israel; 2Assuta Medical Center, Tel-Aviv, Israel, The Sackler School of Medicine, Tel-Aviv University, Tel-Aviv 8436322, Israel; 3Oncology, Kaplan Medical Institute, Rehovot, Hadassah Medical School, The Hebrew University, Jerusalem 9190501, Israel

**Keywords:** *BRCA1/BRCA2*, management, surveillance, guidelines, risk-reducing surgery, early detection scheme

## Abstract

**Simple Summary:**

Female carriers of germline pathogenic/likely pathogenic variants (P/LPVs) in the *BRCA1/BRCA2* (*BRCA*) genes are at a substantially increased lifetime risk for developing breast, ovarian, and (to a lesser extent) other cancer types. Multiple national and international surveillance guidelines and recommendations to facilitate the early detection of cancer in these high-risk women have existed for more than 2 decades. Yet, inconsistencies pertaining to the medical management of cancer-free carriers linger, and surveillance recommendations are not globally harmonized. In this review, we discuss the differences between existing surveillance guidelines for *BRCA* P/LPV carriers, emphasizing the importance of future studies to enable guidelines harmonization and personalized risk stratification for optimal, effective surveillance strategies.

**Abstract:**

Females harboring germline *BRCA1/BRCA2* (*BRCA*) P/LPV are offered a tight surveillance scheme from the age of 25–30 years, aimed at early detection of specific cancer types, in addition to risk-reducing strategies. Multiple national and international surveillance guidelines have been published and updated over the last two decades from geographically diverse countries. We searched for guidelines published between 1 January 2015 and 1 May 2022. Differences between guidelines on issues such as primary prevention, mammography screening in young (<30 years) carriers, MRI screening in carriers above age 65 years, breast imaging (if any) after risk-reducing bilateral mastectomy, during pregnancy, and breastfeeding, and hormone-replacement therapy, are just a few notable examples. Beyond formal guidelines, *BRCA* carriers’ concerns also focus on the timing of risk-reducing surgeries, fertility preservation, management of menopausal symptoms in cancer survivors, and pancreatic cancer surveillance, issues that, for some, there are no data to support evidence-based recommendations. This review discusses these unsettled issues, emphasizing the importance of future studies to enable global guideline harmonization for optimal surveillance strategies. Moreover, it raises the unmet need for personalized risk stratification and surveillance in *BRCA* P/LPV carriers.

## 1. Introduction

Germline pathogenic/likely pathogenic variants (P/LPVs) in the *BRCA1* (OMIM 113705) and *BRCA2* (OMIM 600185) genes were the first clinically significant genetic risk factors identified in families displaying an unusual cluster of breast and ovarian cancer—Hereditary Breast Ovarian cancer syndrome (HBOC) [1,2]. The contribution of these genes (mostly *BRCA2*) to pancreatic cancer [3,4], male breast cancer [5], and prostate cancer [6] susceptibility was also subsequently demonstrated. Recently, a large family-based study suggested that *BRCA2* P/LPVs are associated with a statistically significant risk for stomach cancer, while associations of *BRCA1* P/LPVs carriers with prostate cancer or cutaneous melanoma could not be confirmed [7].

Traditionally, germline *BRCA* genotyping was offered primarily in order to evaluate the carrier’s cancer risks, guide the surveillance scheme, or offer risk-reducing measures [8]. With the expanding role of poly (ADP-ribose) polymerase (PARP) inhibitors as part of the therapeutic armamentarium for early and advanced *BRCA*-associated breast, ovarian, pancreatic, and prostate cancers [9,10], genetic testing has rapidly evolved in order to identify biomarkers predictive of possible therapeutic response. This expansion of individuals offered *BRCA* genotyping is expected to result in increased rates of *BRCA* P/LPV among cancer patients. In addition, wider use of somatic genomic analyses in advanced cancer settings for the possible identification of molecular targets to guide oncological treatment [11,12] will lead to incidental detection of *BRCA* P/LPVs germline carriers. Recent studies in unselected populations demonstrated much higher population prevalence rates of *BRCA* P/LPVs (1:200–1:270) than previously estimated [13,14,15]. These do not include populations such as Ashkenazi Jews with an extremely high prevalence (2.5%) of three predominant PVs due to the founder effect [16]. Thus, population-based founder PV *BRCA* genotyping in specific populations is becoming more acceptable and implemented. Noteworthy, while there is no need for pre-test genetic counseling, anyone who is found to harbor *BRCA* P/LPSV as a result of this screen is provided with a formal oncogenetic counseling session. For example, genotyping for Ashkenazi Jews has been included in the health basket in Israel (and hence covered by the HMOs) since January 2020, and soon similar population-based screens will be offered by the NHS for the Jewish population in the UK. In addition, population-based genetics efforts (e.g., All of Us (https://allofus.nih.gov, accessed on 1 July 2022), the UK biobank (https://www.ukbiobank.ac.uk/, accessed on 1 July 2022)), combined with more accessible and affordable direct-to-consumer genetic testing are all expected to lead to a substantial increase in the rates and numbers of asymptomatic *BRCA* P/LPV carriers worldwide.

This prospect of increasing numbers of cancer-free *BRCA* P/LPVs carriers raises challenges for the health care systems: shortage of genetic counselors, paucity of community-based high-risk clinics and qualified physicians for carrier comprehensive surveillance management, and resource thin MRI, to name a few. Multiple national and international surveillance guidelines have been established, published, and updated since the early 2000s (some of them will be reviewed herein). However, these recommended schemes are variable and differ from each other by country and continent. In this review, we describe and discuss some of the most urgent controversies and open questions relating to the management recommendations for healthy *BRCA* P/LPVs carriers. Hopefully, highlighting the common themes and the differences between existing guidelines may promote international discussion and harmonization efforts.

## 2. Methods

MEDLINE and web-based sources were searched for guidelines updated or published in English (or translated to English) by national and international professional societies or working groups, between 1 January 2015, to 1 May 2022. Guidelines referring to surveillance of healthy (=cancer free; asymptomatic) carriers of germline *BRCA* P/LPVs carriers were included.

From all guidelines identified, multiple aspects of management of healthy *BRCA* P/LPVs carriers were extracted and compared: primary active risk-reducing approaches by modifying lifestyle, chemoprevention, and/or risk-reducing surgeries, early detection screens (secondary prevention), management of menopausal symptoms and effects of lack of endogenous hormones following risk-reducing bilateral oophorectomy (rrBSO) prior to menopause, and fertility preservation.

## 3. Results

Fifteen guidelines were included in this review, and only three of them were published by governmental bodies—the National Institute for Health and Care Excellence (NICE), the French National Cancer Institute (INCa), and the Australian Government eviQ guidelines. The additional 12 guidelines were published by professional bodies. The recommendations in these geographically diverse guidelines are presented in the context of the existing literature. Several guidelines cover only issues related to breast cancer screening, while others focus on the gynecologic management of the carriers.

### 3.1. Primary Prevention

#### 3.1.1. Modifiable Risk Factors

Some modifiable factors of cancer risk in *BRCA* P/LPV carriers have been postulated or assumed by extrapolation from studies of non-carrier women. A meta-analysis of 44 studies published in 2014 focusing on hormonal and exogenous risk factors for breast cancer in *BRCA* P/LPV carriers reported that the only variable displaying a statistically significant association with reduced breast cancer risk is late age at first live birth (>30 years vs. younger and 25–29 years vs. younger) in *BRCA1* (but not in *BRCA2*) P/LPV carriers (Effect Size = 0.65; 95% CI = 0.42 to 0.99) [17]. This trend is in contrast to the observed protective effect of early childbirth in the general non-carrier population [18]. Two hypotheses for this inconsistency were suggested by the authors: (1) the effect of age at first live birth is different in *BRCA1* P/LPV carriers compared with non-carriers, or (2) the possible effect of rrBSO or bias in ascertainment. Although data assessing many additional potential modifiable risk factors did not reach statistical significance, possible associations with increased breast cancer risks were shown for oral contraceptive use (both for *BRCA1* and *BRCA2*) and smoking (*BRCA2*), whereas decreased breast cancer risks were demonstrated with breastfeeding and late age at menarche (*BRCA1*), and decreased ovarian cancer risk with breastfeeding, tubal ligation (*BRCA1*) and oral contraceptives (*BRCA1* and *BRCA2*) [17]. Additional studies focusing on practically modifiable risk factors that may have potential clinical utility are warranted. Of the published guidelines reviewed herein, only the European Society for Medical Oncology (ESMO) [19] and NICE [20] refer to lifestyle modifications as possible measures to be discussed and encouraged with *BRCA* P/LPV carriers.

#### 3.1.2. Chemoprevention

Limited data are available on cancer risk reduction using hormonal modifying agents (e.g., tamoxifen, raloxifene, or aromatase inhibitors) as primary chemoprevention for *BRCA* P/LPVs carriers. In the National Surgical Adjuvant Breast and Bowel Project (NSABP) Breast Cancer Prevention trial (P-1 trial), tamoxifen reduced breast cancer risk by 62% in *BRCA2* P/LPV carriers (relative risk [RR] 0.38, 95% CI 0.06–1.56), but not in *BRCA1* P/LPV carriers (RR 1.67, 95% CI 0.32–10.07), similar to the reduction in the incidence of ER-positive breast cancer among non-carriers in the same P-1 trial [21]. It should be noted that although not statistically significant, this analysis was limited by the small number of *BRCA* P/LPV carriers (8/288 and 11/288 were carriers of a *BRCA1* and a *BRCA2* P/LPV, respectively). In large chemoprevention studies of cancer-free postmenopausal women at increased risk for breast cancer as assigned by family history (but not genotyped for *BRCA* P/LPVs), both raloxifene and aromatase inhibitors (AIs) were associated with a lower risk of breast cancer [22,23,24]. The guidelines by ESMO [19], Spanish Society of Medical Oncology (SEOM) [25], American College of Obstetricians and Gynecologists (ACOG) [26], and National Comprehensive Cancer Network (NCCN) [27] suggest that the use of tamoxifen may be considered, stressing the fact that the level of evidence for this intervention is weak. NICE guidelines do not address this option specifically in *BRCA* P/LPV carriers but do refer to it as an option to be discussed with women at high risk for breast cancer unless they have undergone bilateral risk-reducing mastectomy [20]. The Australian Government eviQ guidelines also suggest considering tamoxifen for pre-/postmenopausal and raloxifene or AIs for postmenopausal *BRCA* P/LPV carriers but recommend that assessment of risks and benefits for the individual woman should be performed by an experienced medical professional [28]. The French National Cancer Institute (INCa) recommends considering hormonal risk-reducing agents for the primary prevention of breast cancer in the *BRCA* P/LPV carriers in the context of clinical trials [29].

Combined hormonal contraceptive (oral contraceptives—OC) use was shown to be an effective method of chemoprevention for ovarian/tubal/peritoneal cancer both for women in the general, average-risk population and for *BRCA* P/LPVs carriers [30,31,32]. Although concerns regarding increased risk for breast cancer with OC use have been raised in case-control studies [33,34], several meta-analyses and systematic reviews found no evidence of significantly increased breast cancer risk in *BRCA* P/LPV carriers [17,32,35], nor in the general population [36]. The conflicting results may be explained by the differences in the specific studies design included in the analyses, such as unmeasured confounding factors in some studies, definitions of duration and timing of exposure, type and dose of OC, or calendar time in which the OC were taken, survival bias and others. The NCCN, ESMO, ACOG, National Society of Genetic Counselors (NSGC), and SEOM suggest considering OC use as a mean of ovarian cancer risk reduction but encourage discussing the risks and benefits [19,25,26,27,37]. The INCa guideline notes that OC may be offered, with rules for OC prescription to be the same for *BRCA* P/LPV carriers as for average-risk women [29]. According to NICE guidelines, women should not be prescribed OC solely for cancer risk reduction or prevention, although in some situations, reduction in ovarian cancer risk may outweigh any possible increase in breast cancer risk. In addition, the NICE guidelines state that only for women with *BRCA1* mutations, the opposing effects of potentially increased risk of breast cancer versus lifetime protection against ovarian cancer by taking OC should be discussed [20]. eviQ guidelines also point out that although there is evidence that combined OC use can reduce ovarian cancer risk, it is significantly less effective than RRSO and not recommended for the sole or even main purpose of cancer prevention [28].

#### 3.1.3. Risk-Reducing Mastectomy

Two meta-analyses concluded that risk-reducing mastectomy (RRM) decreased breast cancer rates, with one of them showing that it also decreased all-cause mortality [38,39]. In both retrospective and prospective observational studies, RRM was shown to decrease breast cancer incidence by 90% or more in carriers of *BRCA* P/LPVs [40,41,42,43,44,45,46]. Heemskerk-Gerritsen et al. found that RRM was associated with lower overall (hazard ratio 0.4) and breast cancer-specific (hazard ratio 0.06) mortality for *BRCA1* P/LPV carriers, but not *BRCA2* P/LPV carriers in a multicenter cohort study [46]. As expected, lower cancer rates would result in lower rates of cancer treatment [47,48], which can be associated with significant morbidity and reduced quality of life. Skin-sparing mastectomy with or without nipple-areolar complex preservation followed by immediate breast reconstruction is becoming the preferred surgical approach given the superior cosmetic results [49,50]. This surgical procedure is considered safe in cancer-free carriers, with no events of new breast cancer, although the follow-up in published studies was relatively short (median 34–42 months). Practically, all guidelines reviewed herein (except for SOGS and ACR, which do not address this issue) recommend raising and discussing RRM as an active risk-reducing option. It is emphasized that discussion of this option should include a detailed elaboration of the risks, benefits (including potentially avoiding breast cancer treatment), and potential negative impact of the procedure on body image and sexuality, as well as consideration of family history, residual breast cancer risk with age, and life expectancy. Only INCa recommends limiting the option for RRM to women who are 30–60 years old and to be assessed on a case-by-case basis over the age of 65 years [29]. The German Working Group on Gynecological Oncology (AGO) states that RRM reduces mortality only in *BRCA1* P/LPV carriers and that RRM counseling should be individualized [51].

#### 3.1.4. Risk-Reducing Oophorectomy

Several studies reported that risk-reducing bilateral salpingo-oophorectomy (rrBSO) reduced ovarian cancer risk and overall mortality [44,52,53,54,55]. A meta-analysis of three prospective studies demonstrated an 80% reduction in ovarian cancer and a 68% reduction in all-cause mortality [56]. Another meta-analysis of 10 studies demonstrated a 79% reduction in the risk for ovarian/fallopian cancer following rrBSO [57]. The role of rrBSO for breast cancer risk reduction has been assessed by multiple studies, mostly reporting reduced risk [52,55,58,59,60,61,62,63]. However, due to methodological issues and additional potential biases of these studies, risk reduction magnitude and its clinical implication are not well-defined. Moreover, evidence is inconsistent regarding the effect of age at rrBSO, and the effect of the specific mutated gene on clinical outcomes of rrBSO. Since *BRCA1* P/LPV carriers tend to develop ovarian cancer at younger ages than *BRCA2* P/LPV carriers [54], most reviewed guidelines addressing this question recommend rrBSO after finalizing family planning between 35 and 40 years of age for *BRCA1* P/LPV carriers, and between 40 and 45 years of age for *BRCA2* P/LPV carriers (Table 1). INCa guidelines recommend rrBSO above age 40 for *BRCA1* and *BRCA2* P/LPV carriers, suggesting that in *BRCA2* P/LPV carriers, the procedure may be deferred to age 45 [29]. NICE guidelines recommend discussing the risks and benefits of the procedure with all known or suspected carriers and deferring it until women have completed their family planning and childbirth [20]. Belgian HBOC guidelines recommend strongly considering rrBSO before age 40 years for *BRCA1*/before 50 years for *BRCA2* P/LPV carriers [64]. ESMO guidelines recommend rrBSO at age 35–40 (regardless of the specific gene mutated) but considering mutation type, the patient’s preferences, and family history [19]. Indian Council of Medical Research recommends offering rrBSO at age 35–40 years regardless of the mutated gene [65].

Notably, none of the guidelines reviewed herein considers salpingectomy without oophorectomy as a standard of care in the absence of safety data from randomized trials.

#### 3.1.5. Other Prophylactic/Risk-Reducing Surgeries

Several studies reported an increased risk for uterine cancer in women with *BRCA* P/LPVs, specifically for serous papillary carcinoma [70,71,72,73,74]; however, the absolute risks were low (between 1.1–4.7%). Recently published data encompassing more than 5000 carrier families found no association with increased uterine cancer risk [7]. Accordingly, none of the guidelines recommends routine hysterectomy in *BRCA* P/LPV carriers. While the procedure is not justified for uterine cancer prevention per se, it should be discussed and individualized in the context of unopposed estrogen replacement therapy in cancer-free women or tamoxifen therapy in breast cancer survivors. This issue is addressed by the Society of Obstetricians and Gynecologists of Canada (SOGC), ACOG, NCCN, National Breast Cancer Consultation of Netherlands (NABON), and eviQ guidelines [26,27,28,67,75].

### 3.2. Secondary Prevention

#### 3.2.1. Breast Cancer Screening

*Physical examination*. Most guidelines call for “breast awareness” from age 18 years and/or clinical breast examination (CBE) every 6–12 months from 25 years of age or starting 5–10 years prior to the age of the youngest breast cancer patient in the family, whichever is earlier (Table 2). However, randomized trials comparing clinical breast exam to no clinical screening have not been published, the impact (if any) of CBE on the detection of cancer is presumably small, and the rationale for recommending CBE is mostly for improved compliance with a comprehensive surveillance program and the concern for tumors appearing before the age of recommended imaging screening or interval cancers [76].

*MRI.* Screening by MRI is recommended by multiple guidelines, as its use has been shown to result in higher detection rates, an earlier stage of disease at diagnosis, and cost-effectiveness [77,78,79,80]. An additional advantage of MRI screening is minimizing radiation exposure. The impact of MRI screening on survival is not clear, despite the stage shift at diagnosis. Although a small retrospective study from Israel suggested a possible survival advantage for *BRCA* P/LPV carriers who decline RRM and develop breast cancer during intensive follow-up compared with unscreened carriers (unaware of their genetic status) [47], a larger study from another tertiary center in Israel did not demonstrate the same positive outcomes [48]. Potential disadvantages for MRI use at annual intervals include higher costs compared with mammography (MG), higher false-positive rates [81], and limited availability in certain geographical regions within the country. Most guidelines recommend starting MRI screening between 25 and 30 years. A recently published study by Boddicker and co-authors [82] showed that *BRCA* P/LPV carriers over age 65 years continue to be at increased risk of breast cancer, with remaining lifetime risk approaching 20%, implying continued MRI screening beyond age 70 years, as suggested by some of the guidelines. However, the optimal surveillance intervals, combining MRI with additional modalities, and the management of younger (<30 years) and older (>70 years) carriers remain controversial and inconsistent among various guidelines, as can be seen in Table 2.

*Mammography*. Screening mammography (MG) has been used as the standard modality for early breast cancer detection for average-risk women over the last few decades, as it demonstrated reduced breast cancer mortality in women older than 40 years [83,84,85,86]. However, there are no data indicating survival advantage with MG as a sole breast imaging modality in *BRCA* P/LPV carriers. The lower sensitivity of MG may be attributed to younger median age at diagnosis, a time in a woman’s life when breast density is high [87], biologically more aggressive tumors [88], and higher rates of interval tumors [89]. Tomosynthesis is not routinely recommended, but extrapolating from average-risk population data, is to be considered according to several of the guidelines (Table 2), given increased cancer detection rates and decreased false-positive recall rates, especially for those younger than 50 years of age [90,91,92]. Nevertheless, Riedl et al. suggested limited added value of MG in MRI-screened carriers regardless of patient age and breast density [93]. Analysis of 3 Canadian studies in carriers undergoing imaging screening suggested consideration of alternative screening protocols below the age of 40 years [94]. Phi et al. also questioned the role of MG in carriers of *BRCA1* P/LPVs under the age of 40 years [95]. While there are remarkable differences between the surveyed guideline in the screening recommendations regarding MG use (Table 2), at this stage, only NABON guidelines distinguish between the recommended imaging surveillance scheme for *BRCA1* and *BRCA2* P/LPV carriers [68], while the American College of Radiology (ACR), Austrian Clinical Practice Guideline, and SEOM guidelines suggest considering/discussing delaying MG until age 40 years for *BRCA1* P/LPV carriers undergoing annual MRI screening [25,66,69]. Most guidelines recommend discontinuing MG screening at 75 years, as in the general population, or considering it on an individual basis.

*Breast Ultrasound*. Whole-breast ultrasound (US) was shown to increase the detection rate of clinically favorable cancers, but this was accompanied by an increased rate of false positive findings [96,97]. In addition, when screening involved MRI and MG, there was no added value for US screening [93,98,99]. Most guidelines agree that US should only be used as a supplementary method to MG when MRI is not available/suitable or is difficult to interpret (Table 2).

*Surveillance during pregnancy and breastfeeding*. The incidence of pregnancy-associated breast cancer is low in the general population (~1 in every 3000 pregnancies) [100]. Recently published single institution experience found a 2.05% detection rate of breast cancer in *BRCA* P/LPV carriers screened by breast US during 7 years follow-up [101]. This low incidence may underlie the paucity of data in the literature referring to the recommended surveillance scheme during pregnancy or postpartum, specifically in *BRCA* P/LPV carriers [102]. Only 3 of the guidelines reviewed here address this issue (NABON, eviQ, and Austrian Clinical Practice Guideline), and all suggest considering clinical examination or US follow-up (Table 2) [28,68,69].

*Surveillance following RRM*. Since RRM significantly reduces the risk of breast cancer, with an absolute risk of up to 1.5% at follow-up ranging between 2 to 13 years [103], none of the guidelines recommends continued imaging surveillance after RRM. ESMO guidelines suggest considering annual breast MRI or ultrasound after nipple-sparing mastectomy because of the concern of a higher risk of cancer with more residual tissue [19]; however, no long-term safety data for this procedure exist. The Austrian Clinical Practice Guideline states that an annual MRI can be offered to healthy carriers women and should be offered for a follow-up to carriers previously diagnosed with breast cancer [69].

*Breast cancer screening in male carriers*. The cumulative male breast cancer risk is estimated at 0.1–1.5% in *BRCA1* P/LPV carriers and at 1.9–7.7% in *BRCA2* P/LPV carriers [7]. Male surveillance is addressed by the NCCN (from age 35 years), ESMO (from age 30 years), Belgian HBOC guidelines (from age 40 years), SEOM, AGO, eviQ, and Austrian Clinical Practice Guideline (the latter four do not specify an age for commencing screening) [19,25,27,28,51,64,69]. Belgian HBOC guidelines suggest surveillance only for *BRCA2* P/LPV carriers (considering annual clinical exam by physician), with no routine screening for *BRCA1* carriers [64]. All others suggest the same surveillance regardless of the mutated gene, with breast awareness and self-examination recommended by all, clinical exam recommended by NCCN and ESMO, and considering MG in the case of gynecomastia (NCCN—starting at age 50 years or 10 years before the earliest known male breast cancer in the family, and SEOM) [19,27] and for a suspicious lesion noted on clinical or self-performed breast exam (Austrian guidelines) [69].

#### 3.2.2. Ovarian Cancer Screening

Multimodality screening with transvaginal ultrasound (TVUS) and CA-125 versus either TVUS alone or no screening showed that multimodality screening is more effective at detecting early-stage cancer [104,105,106]. In addition, Skates et al. evaluated the risk of ovarian cancer algorithm (ROCA) by following increase in CA-125 above each subject’s baseline q3 months, which triggered TVUS. This study found that, although without an unscreened control group but rather compared to the historical controls, it had better sensitivity for early-stage diagnosis, high specificity, and low yet possibly acceptable PPV compared with CA125 > 35 U/mL q6/q12 months [107]. However, no survival advantage in screened patients was observed with longer follow-ups in the randomized studies. While NCCN, ESMO, ACOG, and SEOM guidelines suggest considering screening from age 30 years, acknowledging its limited value [19,25,26,27], other guidelines do not recommend any type of surveillance aimed at the early detection of ovarian cancer (Table 1). None recommends continued screening for primary peritoneal carcinoma because of the small residual risk following rrBSO (estimated cumulative incidence of 4.3% at 20 years after oophorectomy) [108].

#### 3.2.3. Pancreatic Cancer Screening

Two studies on pancreatic ductal adenocarcinoma (PDAC) surveillance outcomes in genetically predisposed high-risk individuals have been published recently with conflicting results. The Dutch Familial Pancreatic Cancer Surveillance Study Group did not find survival advantage for annual screening with both endoscopic ultrasonography (EUS) and MRI/cholangiopancreatography (MRCP) [109]. In this cohort, 165 patients had genetic predisposition (45 of them had *BRCA2* P/LPVs and 7 had *BRCA1* P/LPVs), and the average follow-up was 63 months. By contrast, the analysis of the multicenter Cancer of Pancreas Screening (CAPS5) study, which included 269 *BRCA2* P/LPV carriers and 68 *BRCA1* P/LPV carriers with a median follow-up of 4 years, found that 77.8% of the PDACs diagnosed within the high-risk cohort (screened by annual EUS and/or MRI/MRCP, with modified surveillance interval in case of radiologically suspicious lesion) had stage I disease and 5-year survival of 73.3%. This is compared with the 5-year overall survival of 11% in patients with PDAC in the US [110]. The differences in outcome between these two studies could be attributed to the number of non-*BRCA* P/LPV carriers: in the Dutch study, 58% carried a P/LPV in the *CDKN2A* gene, whereas in the CAPS5, less than 5% carried P/LPV in the same gene, and *CDKN2A*-associated PDAC are considered to have a more aggressive disease course. Until more robust data are published, specifically in *BRCA* P/LPV carriers, there is no consensus in the reviewed guidelines regarding optimal PDAC surveillance, especially given the low lifetime risk for PDAC in *BRCA* P/LPV carriers, estimated at around 5% [7,111], and questioning the cost-effectiveness of such surveillance scheme. The 2022 American Society for Gastrointestinal Endoscopy (ASGE) guideline on screening for PDAC in patients with genetic susceptibility made a conditional recommendation for PDAC screening in *BRCA* P/LPV carriers regardless of family history (unlike in previous guidelines), although stating that this recommendation has very low quality of evidence [111]. The guidelines reviewed here refer to older studies, with only NCCN, ESMO, NABON, SEOM, NICE, eviQ, and the Belgian guidelines addressing this aspect of screening (Table 3) [19,20,25,27,28,64,68]. ESMO guidelines note that data supporting screening are very limited, based on a study from 2013 [112], and encourage participation in clinical trials assessing the efficacy of screening techniques for PDAC [19]. NABON, eviQ, and Belgian guidelines recommend screening for PDAC only as part of clinical trials [28,64,68].

#### 3.2.4. Prostate Cancer Screening

*BRCA*-related prostate cancer was shown in several studies to be more aggressive and affects men at a younger age than sporadic prostate cancer [113,114,115]. Indeed, in the context of the IMPACT study, *BRCA2* P/LPV carriers (n = 902) had a higher incidence of prostate cancer, younger age of diagnosis, and clinically significant tumors compared with non-carriers (n = 497) [115]. These data prompted the authors to recommend systematic PSA for men harboring *BRCA2* P/LPVs. Knowledge of germline mutational status during screening is considered a factor in a decision regarding surgery versus active surveillance [116]. The seemingly elevated prostate cancer risk for *BRCA1* P/LPV carriers was not confirmed in a recent family-based cohort [7] that was published after the updated versions of the reviewed guidelines. Several guidelines address prostate cancer screening, with minor, clinically negligible differences (Table 3).

#### 3.2.5. Other Cancer Types

A recent large retrospective, family-based study suggested a small, but statistically significant, increased risk for gastric cancer in *BRCA* P/LPV carriers (RR = 2.17 for *BRCA1* and RR = 3.69 for *BRCA2*), slightly increased risk for biliary tract cancers in *BRCA1* P/LPV carriers (RR = 3.34), and for colorectal cancer in *BRCA1* P/LPV carriers (RR = 1.48) [7]. Previously suggested increased risks for melanoma and endometrial cancer were not demonstrated in the same study. NCCN, ESMO, and SEOM address screening for some of these malignancies, based on older literature (Table 3) [19,25,27].

### 3.3. Other Aspects of Management of Healthy Carriers

#### 3.3.1. Hormone Replacement Therapy after rrBSO

The use of Hormone Replacement Therapy (HRT) for the management of menopausal symptoms in carriers after rrBSO is controversial since data from the few available studies are conflicting. Kotsopoulos et al. did not find an increased risk for breast cancer in *BRCA1* P/LPV carriers treated with HRT, but in those taking combined (estrogen plus progesterone) HRT, the 10-year actuarial risk of breast cancer was 22%, as compared with 12% in those taking estrogen alone, with no statistically significant differences based on age [117]. The PROSE Study Group did not demonstrate increased breast cancer risk with short-term HRT use (mean follow-up of 2.6 years, range of 0.1–19 years) [118]. In a recently published retrospective cohort of 306 consecutive healthy carriers, patients older than 45 years had higher rates of breast cancer (21% vs. 8%) compared to those who did not use HRT post-rrBSO; however, no increased risk was observed in patients younger than 45 years [119]. The guidelines addressing this issue vary, with some recommending offering HRT while others suggest considering and discussing risks and benefits with the individual carrier, ultimately making the decision (Table 1) and thoroughly summarized by Manchanda et al. [120].

#### 3.3.2. Vaginal Estrogen Therapy

Genitourinary syndrome of menopause, which includes dyspareunia and vaginal dryness, is frequent after rrBSO [121]. There are no data regarding the risk of breast cancer in high-risk women treated with vaginal estrogen therapy. SOGC guidelines consider this option safe in cancer-free carriers (although they recommend non-hormonal alternatives first in women with a personal history of breast cancer) [67].

#### 3.3.3. Contraception

Other than the use of OC as a measure of risk-reduction for ovarian cancer (discussed above), the issue of safe contraception should be discussed with young carriers. Besides recommendations described in the chemoprevention section and Table 1, NABON interpret the data in the literature cautiously and state that oral contraceptives lead to a transient slightly increased RR for breast cancer (RR = 1.25) during their usage, but the absolute risk increase for breast cancer is very limited if used before the age of 25, given the very low absolute risk at that age (including in carriers) [68]. However, they do not address the issue of oral contraceptives above the age of 25 and state that there are no data about the safety of levonorgestrel-containing intra-uterine devices (IUD) as a contraception method for *BRCA* P/LPVs carriers. They also recommend considering the fact that cycle-guided breast MRI for screening may be more difficult to plan and interpret in women who experience levonorgestrel-containing IUD-associated amenorrhea [68]. eviQ guidelines point out that the effect of oral contraceptives on breast cancer remains unclear, and where an adverse effect on breast cancer risk is reported, the effect is small and similar to that reported in the general population [28]. SOGC point out that OC use in young *BRCA1* carriers should be individualized, taking into account the risks and benefits [67], while ACOG state that given the magnitude of the potential benefits (e.g., ovarian and endometrial cancer risk reduction, pregnancy prevention, cycle regulation), OC use in carriers is appropriate if indicated, and use for cancer risk reduction is reasonable [26].

#### 3.3.4. Effects of In Vitro Fertilization and Pre-Implantation Genetic Diagnosis

The safety of in vitro fertilization (IVF) in *BRCA* P/LPV carriers was assessed in 2008 by a group from Toronto’s Women’s College Hospital. That study reported that neither a history of infertility and the use of fertility medications, nor IVF were associated with an increased breast cancer risk [122]. Data published later from HEBON also found no evidence of increased breast cancer risk with ovarian stimulation for IVF in *BRCA* P/LPV carriers [123]. A historical cohort study on 1824 Jewish Israeli *BRCA* P/LPV carriers reported no association between fertility treatments (ovarian stimulation, IVF, or their combination) and increased breast cancer risk [124]. A matched case-control study in carriers with and without ovarian cancer did not find an association between IVF and ovarian cancer risk [125]. SOGC guidelines state that carriers affected by infertility can safely undergo fertility treatments [68]. Additional guidelines, including NCCN, ESMO, Belgian Society for Human Genetics, NABON, NSGC, INCa recommend discussing the options of fertility preservation and pre-implantation genetic diagnosis with the carriers [19,27,29,37,64,68].

#### 3.3.5. Effect of Specific Family History on Recommendations

As shown in Table 1, Table 2 and Table 3, most guidelines recommend considering the family history of cancer and ages at diagnosis of cancer in the family as factors for decision making. These refer mostly to the age of starting various screening procedures but less to the age recommended for risk-reducing surgeries (and specifically rrBSO), where patient’s preferences and family planning status should be considered and may be more dominant factors in making these decisions than the family history.

## 4. Discussion

This review depicts multiple aspects of management for cancer-free carriers of germline P/LPVs in *BRCA1* and *BRCA2* genes, including risk-reducing options, secondary prevention (screening), as well as treatment of iatrogenic sequela of risk-reducing interventions. Controversies and inter-guidelines inconsistencies pertaining to the medical management of the carriers exist. This variability may be attributed to a different interpretation of the data presented in the published literature, insufficient levels of evidence for certain recommendations, or possibly national specific cost–benefit analyses, as well as international and cultural variations in the perception of risk and the acceptance of risk-reducing measures. These inconsistencies are mostly notable in breast imaging, including surveillance intervals, combining MRI with additional imaging modalities, and screening of younger (<30 years) and older (>60 years) carriers. Some of these inconsistencies may be explained by rapidly changing data regarding risk assessment, paucity of robust data from prospective randomized trials, literature interpretation, frequency of guideline updates, and cost-effectiveness analyses used by each group/society. These controversies may account for the fact that some medical centers use self-developed protocols rather than national guidelines for carriers’ surveillance [126]. These clinically relevant inconsistencies also emphasize the importance of international studies to enable global guideline harmonization for optimal surveillance strategies and appropriate counseling for newly diagnosed carriers. Moreover, it raises the unmet need for personalized risk stratification and surveillance in *BRCA* P/LPV carriers.

Since screening for cancer does not actively reduce cancer risk but only offers a passive way to a possible earlier stage at diagnosis (with an unclear effect (if any) on survival), in a sizeable number of cases having to endure morbid oncological treatments [48], the importance of new risk-reducing strategies is self-evident. The existing options of rrBSO in premenopausal women and RRM have significant psychological and quality-of-life impacts, some of which are not optimally addressed. Ongoing studies assessing the safety of additional risk-reducing options, such as salpingectomy with delayed oophorectomy (NCT01907789) and the BRCA-P trial studying the effect of denosumab on breast cancer risk in *BRCA1* P/LPV carriers (NCT04711109), may offer more options with a potentially lower rate of deleterious effects on quality of life in the future. In addition, personalized risk stratification should be investigated by international consortia to enable personalized “next generation” surveillance since this may not only save multiple unnecessary screening procedures for lower-risk carriers and cost for healthcare systems but also encourage higher-risk carriers to consider risk-reducing strategies instead of (hopefully) early detection of cancer. Therefore, when available, women should be encouraged to participate in relevant clinical trials.

## 5. Conclusions

Multiple controversies pertaining to the medical management of healthy *BRCA1* and *BRCA2* P/LPV carriers result in varying recommendations by international societies. Updated international consensus guidelines are warranted for synchronized counseling and surveillance strategies. In parallel, it is important to educate primary care physicians and discuss with the patients the risks and benefits of primary prevention, risk-reducing strategies, and limitations of screening procedures to enable informed and shared decision making.

## Figures and Tables

**Table 1 cancers-14-04592-t001:** Ovarian cancer risk management.

Guidelines	Surveillance before rrBSO TVUS + CA125	Recommended Age for rrBSO—BRCA1	Recommended Age for rrBSO—BRCA2	Surveillance Following rrBSO	HRT Following rrBSO	Other	Ref
NCCN (2022) and NSGC (2021)	Maybe considered starting at 30–35 ^a^	35–40	40–45	NA	Should discuss risks and benefits ^b^	Possible benefit of rrBSO on breast cancer risk, conflicting evidence	[27,37]
ACR (2018)	NA	NA	NA	NA	NA	NA	[66]
SOGC (2018)	Insufficient data to support	35–40	40–45	Not recommended	Should be offered until the average age of menopause ^c^	rrBSO should be considered for breast cancer risk reduction in BRCA2 mutation carriers < 50 years	[67]
ACOG (2017)	Not recommended, may be considered starting at 30–35 until rrBSO	35–40	40–45	Not recommended	Should be offered short-term. Long-term effect on breast cancer risk unknown	Use of OC for ovarian cancer prophylaxis is	[26]
						reasonable	
NICE (Great Britain) (updated 2019)	NA	NA ^d^	NA ^d^	NA	Offer up until the time of expected natural menopause ^e^		[20]
ESMO (2016)	May be considered starting at 30 ^f^	35–40 ^g^	35–40 ^g^	Not recommended	Short-term use is safe among healthy carriers	Conflicting data regarding rrBSO effect on breast cancer risk	[19]
NABON (Netherlands) (updated 2017)	Proved ineffective ^h^	35–40 ^i^	40–45 ^i^	Not recommended	Should be discussed		[68]
INCa (France) (updated 2017)	Annual pelvic clinical examination only	>40	Can be deferred to 45	Not recommended ^k^	Discuss if rrBSO performed before 45 years		[29]
SEOM (Spain) -2020	Consider from age 30 until rrBSO or for those who have not elected rrBSO	35–40	40–45	NA	May be considered, short-term and low-dose	rrBSO for breast cancer reduction should be recommended only to women under the age of 50	[25]
Belgian Society for Human Genetics (updated 2022)	Not recommended	Strongly consider < 40 years	Strongly consider < 50 years	NA	NA		[64]
AGO (updated 2022)	NA	>35 ^g^	>40 ^g^	NA	NA		[51]
Austrian Clinical Practice Guideline -2015	Annual	NA	NA	Not indicated	NA		[69]
Australia (and New Zealand)—Cancer Institute eviQ (updated 2022)	Do not offer	>35 ^i^	>40 ^i^	NA	NA		[28]
Indian Council of Medical Research -2016	Not routinely recommended	35–40	35–40	NA	NA		[65]

ACOG: American College of Obstetricians and Gynecologists; ACR: American College of Radiology; AGO: Arbeitsgemeinschaft Gynäkologische Onkologie; ESMO: European Society for Medical Oncology; HRT: Hormone replacement therapy; INCa: Institut National du Cancer; MG: Mammography; MRI: magnetic resonance imaging; NA: Not addressed; NABON: Nationaal Borstkanker Overleg Nederland; NCCN: National Comprehensive Cancer Network; NICE: National Institute for Health and Care Excellence; NSM: Nipple-sparing Mastectomy; OC: Oral Contraceptives; rrBSO: risk-reducing bilateral salpingo-oophorectomy; SEOM: Sociedad Espanola de Oncologia Médica; SOGC: Society of Obstetricians and Gynaecologists of Canada. US: Ultrasonography. ^a^ At the clinician’s discretion. ^b^ Given the limitations inherent in nonrandomized studies. ^c^ In the absence of contraindications. ^d^ Discuss and include in the discussion the positive effects of reducing the risk of breast and ovarian cancer and the negative effects of surgically induced menopause. Defer risk-reducing bilateral salpingo-oophorectomy until women have completed their family. ^e^ Combined HRT if they have a uterus/estrogen-only HRT if they don’t have a uterus. ^f^ The limited value of these tools as an effective screening measure should be communicated to individuals. ^g^ Mutation type, the patient’s preferences, family planning status, and family history should be taken into consideration when deciding on the age for RRSO. ^h^ Screening can, therefore, only be aimed at early detection of carcinoma. ^i^ After family completion. ^k^ Usual gynecological clinical monitoring if intact uterus.

**Table 2 cancers-14-04592-t002:** Breast cancer screening guidelines.

Guidelines	Breast Exam Start Age–End Age	Annual MRI Start Age–End Age	Breast US Start Age–End Age	MG Start Age–End Age	Surveillance during Pregnancy and Breastfeeding	Following RRM	Ref
NCCN (2022) and NSGC (2021)	25–NA	25 ^d^–75	NA	30 ^e^–75 ^j^	NA	NA	[27,37]
ACR (2018)	NA	25 to 30–NA	When MRI unavailable	30 ^e,f^–NA	NA	NA	[66]
ACOG (2018)	25–NA Every 6–12 months	25–NA	NA	30–NA	NA	NA	[26]
NICE (Great Britain) (updated 2019)	Breast awareness	30–49 50–69 only if dense breast	When MRI is not suitable or when results of MG or MRI are difficult to interpret	Consider 30–39 Offer 40–>70	NA	Surveillance should not be offered	[20]
ESMO (2016)	25 ^a^–NA Every 6–12 months	25–NA	>25 only if MRI unavailable	30–NA	NA	Consider annual breast MRI or ultrasound after NSM	[19]
NABON (Netherlands) (updated 2017)	25–75 annually	25–60 60–75 ^h^	No	40–60 every 2 years (BRCA1) 30–60 annual (BRCA2) 60–75 annual (BRCA1/2) ^h^	Self-examination and clinical examination every 6 m ^i^	Imaging surveillance is not indicated	[68]
INCa (France) (updated 2017)	<30 annually	30 ^d^–65	When clinically indicated	30–>65 (considering comorbidities and life expectancy)	NA	Annual clinical monitoring; no imaging surveillance	[29]
SEOM (Spain) -2020	NA	30 ^d^–70	When MRI unavailable	30 ^f^–75	NA	NA	[25]
Belgian Society for Human Genetics (updated 2022)	25 ^b^–NA semiannual	25 ^b^–65	When results of MRI are difficult to interpret	35 ^g^–75 annual >75 consider every 2 years		No standard follow-up with imaging	[64]
AGO (Germany) (updated 2022)	25–NA Semiannually	25–NA	25–NA	40–NA biannually	NA	NA	[51]
Austrian Clinical Practice Guideline -2015	NA	25 ^b^–NA	When MRI unavailable	35–NA	US in 3-monthly intervals; MRI not earlier than 2 m after lactation has ceased	Annual MRI examinations can be offered ^k^	[69]
Australia (and New Zealand ^L^)—Cancer Institute eviQ (updated 2022)	Breast awareness	30 ^c^–50 >50 consider if dense breast	Consider	40–>50	Consider US	Self-surveillance of breast area	[28]
Indian Counsyl of Medical Research -2016	25 ^b^–NA semiannually	25–NA	Poor sensitivity, ages NA	Poor sensitivity, ages NA	NA	NA	[65]

ACOG: American College of Obstetricians and Gynecologists; ACR: American College of Radiology; AGO: Arbeitsgemeinschaft Gynäkologische Onkologie; ESMO: European Society for Medical Oncology; INCa: Institut National du Cancer; MG: Mammography; MRI: magnetic resonance imaging; NA: Not addressed; NABON: Nationaal Borstkanker Overleg Nederland; NCCN: National Comprehensive Cancer Network; NICE: National Institute for Health and Care Excellence; NSM: Nipple-sparing Mastectomy; SEOM: Sociedad Espanola de Oncologia Médica; US: Ultrasound. ^a^ Or starting 10 years earlier than youngest breast cancer diagnosis in the family. ^b^ Or starting 5 years earlier than the youngest breast cancer diagnosis in the family. ^c^ Or individualized based on family history if a breast cancer diagnosis is present before age 35. ^d^ Or starting earlier if there is a family history of breast cancer before 30 years. ^e^ Considering breast thomosynthesis. ^f^ Discuss delaying mammography until 40 years with *BRCA1* carriers who undergo annual MRI screening. ^g^ Mammography at age 30, annual mammography from 30 onwards in case of microcalcifications. ^h^ In case of heterogeneous dense or very dense fibroglandular tissue (ACR 3 or 4), alternating mammography and MRI (as annual imaging) is advised. ^i^ Women planning to become pregnant are advised to have an MRI every 6 m until they are pregnant. ^j^ Management should be considered on an individual basis over age 75. ^k^ Should be offered for follow-up in women who have a history of breast cancer. ^L^ New Zealand guidelines refer to eviQ recommendations.

**Table 3 cancers-14-04592-t003:** Non-breast/ovarian cancer screening guidelines.

Guidelines	Pancreatic Cancer	Prostate Cancer	Other Cancers	Ref
NCCN (2022)	Consider when ≥1 first- or second-degree relatives with PDAC (*BRCA1* and *2*), from age 50 ^a^	From age 40, recommend for *BRCA2* Consider for *BRCA1*	General risk management for melanoma is appropriate	[27]
NICE (updated 2019)	Offer when ≥1 first-degree relatives with PDAC (*BRCA1* and *2*)	NA	NA	[20]
ESMO (2016)	Consider (*BRCA2* only), from age 50 ^a^	May be considered from age 40, particularly for *BRCA2*	The association with elevated risk of gastric cancer, colorectal cancer, and uterine cancers remains weak, thus screening and prevention generally not indicated	[19]
NABON (Netherlands) (updated 2017)	Offer only in a study context for carriers with ≥2 relatives with PDAC (*BRCA1* and *2*)	NA	NA	[68]
SEOM (Spain) -2020	Consider when 1 first-degree relative with PDAC (*BRCA1* and *2*), from age 50 ^a^	Annual PSA, from age 40—recommend for *BRCA2* Consider for *BRCA1*	Consider skin and eye examination for melanoma screening according to personal/familiar risk factors	[25]
Belgian Society for Human Genetics (updated 2022)	Preferentially in clinical trials, with ≥ 1 first-degree relatives with PDAC (*BRCA1*)/≥1 first-degree relative or ≥ 2 relatives of any degree with PDAC (*BRCA2*)	Annual PSA and DRE from age 50 ^a^ (*BRCA1*)/from age 40 ^a^ (*BRCA2*)		[64]
AGO (updated 2022)	NA	As part of standard care	NA	[51]
Austrian Clinical Practice Guideline (2015)	NA	As part of standard care	NA	[69]
Australia (and New Zealand)—Cancer Institute eviQ (updated 2022)	Lack of evidence of benefit from screening. Should be undertaken only as part of a clinical trial	Consider annual PSA +/− DRE from age 40. If persistent elevation of PSA above normal, refer to a urologist	NA	[28]

AGO: Arbeitsgemeinschaft Gynäkologische Onkologie; DRE: digital rectal exam; ESMO: European Society for Medical Oncology; HRT: Hormone replacement therapy; NA: Not addressed; NABON: Nationaal Borstkanker Overleg Nederland; NCCN: National Comprehensive Cancer Network; NICE: National Institute for Health and Care Excellence; NSM: Nipple-sparing Mastectomy; PDAC: pancreatic ductal adenocarcinoma; rrBSO: risk-reducing bilateral salpingo-oophorectomy; SEOM: Sociedad Espanola de Oncologia Médica; US: Ultrasonography. ^a^ or 10 years younger than the earliest diagnosis in the family, whichever comes first.

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
