# Peer review of "Controversies and Open Questions in Management of Cancer-Free Carriers of Germline Pathogenic Variants in BRCA1/BRCA2"

_cancers, 2022, doi:10.3390/cancers14194592_

Round 1

Reviewer 1 Report

The aim of this study was to describe clinical practice guidelines for management of carriers of pathogenic/likely pathogenic variants in the BRCA1 or BRCA2 genes. The authors provide a summary of international guidelines for risk management for both male and female carriers, highlighting the differences for several risk management interventions as well as the need for more evidence regarding effectiveness.

General comments

1.      The authors comprehensively describe the different guidelines as they apply to management of hereditary cancer risks associated with BRCA1/2 P/LPVs. They also provide context for the guidelines by including descriptions of existing literature for each intervention. For the latter, I think this review could be strengthened by a more critical interpretation of the literature throughout. As an example, for RRBSO (pages 4/5) the authors mention methodological issues and potential biases in a very general way rather than highlighting specific issues/biases and how these impact on outcomes. A particular issue with estimates of the effectiveness of RRBSO for ovarian cancer risk-reduction is that earlier studies in particular did not necessarily have a standardised pathological examination of the surgical tissue (using SEE-FIM protocol), so cancers detected after RRBSO may have already been present at the time of surgery. Another example is page 4 line 162 – what specific aspects of the study design would explain conflicting results?

2.      For the results, it would be helpful to include a brief summary of the guidelines include such as the countries/jurisdictions, the source (e.g. government body vs professional body) and type of healthcare systems. For example a government organisation might place more weight on aspects like cost-effectiveness when preparing guidelines compared to a professional medical organisation.

3.      Discussion. Could any of the variation in guidelines be due to cultural differences? Also, much of the data guidelines rely on are obtained using cohorts that are predominantly of European ancestry.

4.      Discussion. The authors comment on the need for personalised risk stratification and surveillance. In this context due to the increase in multi-gene panel testing it would worth including a discussion of management guidelines for other HBOC genes such as PALB2 and CHEK2, especially as some guidelines already address these (e.g. NCCN, eviQ). Polygenic risk is also relevant to mention for personalised risk stratification, as the combined effect of SNPs can significantly modify penetrance in carriers of monogenic P/LPVs (Barnes et al 2020; Gallagher et al 2020).

Specific comments

1.      Please change all references to the NSW guidelines to eviQ. Although originally for NSW, the eviQ program includes risk management guidelines that are now developed and used nationally in Australia.

2.      Page 2 line 68. I would suggest changing the wording of “genetic counselling-free” here – while population-based testing would mean pre-test genetic counselling is unlikely, individuals who are found to carry a P/LPV would benefit from genetic counselling post-testing (especially given the complexity of management as highlighted in this review).

3.      Page 4 risk-reducing mastectomy. Another benefit of RRM is potentially avoiding breast cancer treatment, which can be associated with significant morbidity and reduced quality of life.

4.      Page 11 Prostate cancer screening. It is worth mentioning the IMPACT study, which is a multi-centre international study investigating targeted PSA screening in BRCA1/2 carriers (Page et al 2019 Eur Urol).

Reviewer 2 Report

This is a review that discusses the differences between existing surveillance guidelines for healthy BRCA-mutation carriers, covering many aspects of their management and including screening of several types of cancers (not only ovarian or breast) and fertility preservation. 

The work focuses on a critical topic since surveillance recommendations are not globally harmonized and thus inconsistencies in the medical approach to these patients persist. 

An exhaustive extraction and comparison of several guidelines updated or published in English (or translated to English) by national and international professional societies or working groups, by May 2022 was performed.  Besides plausible explanations for any differences were provided. 

This work is potentially helpful because a thorough knowledge of the guidelines can guide a more precise counseling to patients, an individualized medicine and a tailored approach. However, some flaws make the paper unpublishable in the present form.

-The introduction is wide and exhaustive. I only assumed that this section is quite long. Thus, I suggest thinning out and summarizing the text and making it more fluent: in a briefer introduction, the authors should focus on how shortcomings or gaps make the paper appropriate and necessary and on the advantages its reading may offer. 

Line 86-89 I think these reflections are more appropriate to be moved into the discussion section. 

-The methods section is quite good but I think some information should be added and I would recommend following the PRISMA guidelines for writing reviews.

-The result section is broad and in-depth but not always easy to be followed and the manuscript, in terms of fluency, should be improved. Furthermore, the division into paragraphs and subparagraphs is a bit chaotic and could be more schematic and consistent.

-The discussion is wide and focuses on fundamental points. 

I suggest outlining a synthetic algorithm/schema/summary with treatment strategies that the authors deem more appropriate or with their clinical practice to share with readers. 

-Tables 1-2-3 are helpful and consistent. I just suggest adding the year of publication near to the name of the guideline for thoroughness.

-The English language is good, with just a few oversights.

-Research In Context is quite good. Nevertheless, I think that some other literature is worth to be mentioned. I suggest including these articles, if the authors agree:

       Kotsopoulos J. BRCA Mutations and Breast Cancer Prevention. Cancers (Basel). 2018 Dec 19;10(12):524. doi: 10.3390/cancers10120524. 

       De Felice F, Marchetti C, Boccia SM, Romito A, Sassu CM, Porpora MG, Muzii L, Tombolini V, Benedetti Panici P. Risk-reducing salpingo-oophorectomy in BRCA1 and BRCA2 mutated patients: An evidence-based approach on what women should know. Cancer Treat Rev. 2017 Dec;61:1-5. doi: 10.1016/j.ctrv.2017.09.005. Epub 2017 Sep 28. PMID: 29028552.

       Manchanda R, Gaba F, Talaulikar V, Pundir J, Gessler S, Davies M, Menon U; Royal College of Obstetricians and Gynaecologists. Risk-Reducing Salpingo-Oophorectomy and the Use of Hormone Replacement Therapy Below the Age of Natural Menopause: Scientific Impact Paper No. 66 October 2021: Scientific Impact Paper No. 66. BJOG. 2022 Jan;129(1):e16-e34. doi: 10.1111/1471-0528.16896. Epub 2021 Oct 20. PMID: 34672090.

       Skates SJ, Greene MH, Buys SS, Mai PL, Brown P, Piedmonte M, Rodriguez G, Schorge JO, Sherman M, Daly MB, Rutherford T, Brewster WR, O'Malley DM, Partridge E, Boggess J, Drescher CW, Isaacs C, Berchuck A, Domchek S, Davidson SA, Edwards R, Elg SA, Wakeley K, Phillips KA, Armstrong D, Horowitz I, Fabian CJ, Walker J, Sluss PM, Welch W, Minasian L, Horick NK, Kasten CH, Nayfield S, Alberts D, Finkelstein DM, Lu KH. Early Detection of Ovarian Cancer using the Risk of Ovarian Cancer Algorithm with Frequent CA125 Testing in Women at Increased Familial Risk - Combined Results from Two Screening Trials. Clin Cancer Res. 2017 Jul 15;23(14):3628-3637. doi: 10.1158/1078-0432.CCR-15-2750. Epub 2017 Jan 31. PMID: 28143870; PMCID: PMC5726402.

       Elezaby M, Lees B, Maturen KE, Barroilhet L, Wisinski KB, Schrager S, Wilke LG, Sadowski E. BRCA Mutation Carriers: Breast and Ovarian Cancer Screening Guidelines and Imaging Considerations. Radiology. 2019 Jun;291(3):554-569. doi: 10.1148/radiol.2019181814. Epub 2019 Apr 30. PMID: 31038410.
